# Morphological evolution of language-relevant brain areas

**Guillermo Gallardo**[1]*, **Cornelius Eichner**[1], **Chet C. Sherwood**[2], **William D. Hopkins**[3], **Alfred Anwander**[1], **Angela D. Friederici**[1]

**1** Department of Neuropsychology, Max Planck Institute for Human Cognitive and Brain Sciences, Leipzig, Germany, **2** Department of Anthropology, The George Washington University, Washington, DC, United States of America, **3** Department of Comparative Medicine, Michale E. Keeling Center for Comparative Medicine, The University of Texas MD Anderson Cancer Center, Bastrop, Texas, United States of America

◔ These authors contributed equally to this work.

* gallardo@cbs.mpg.de

**Data Availability Statement:** The probabilistic chimpanzee atlases of BA44 and 45, alongside the respective processing scripts used to generate them are publicly available for download in our repository: https://github.com/gagdiez/

## Abstract

Human language is supported by a cortical network involving Broca's area, which comprises Brodmann Areas 44 and 45 (BA44 and BA45). While cytoarchitectonic homolog areas have been identified in nonhuman primates, it remains unknown how these regions evolved to support human language. Here, we use histological data and advanced cortical registration methods to precisely compare the morphology of BA44 and BA45 in humans and chimpanzees. We found a general expansion of Broca's areas in humans, with the left BA44 enlarging the most, growing anteriorly into a region known to process syntax. Together with recent functional and receptorarchitectural studies, our findings support the conclusion that BA44 evolved from an action-related region to a bipartite system, with a posterior portion supporting action and an anterior portion supporting syntactic processes. Our findings add novel insights to the longstanding debate on the relationship between language and action, and the evolution of Broca's area.

## Introduction

Language processing is a human trait that recruits Broca's area in the inferior frontal gyrus [1–3]. Previous studies suggest an involvement of this area in the understanding and imitation of actions [4]. Moreover, homologous areas in nonhuman primates have similarly been shown to support actions of the orofacial muscles and upper limbs [5]. Despite extensive research, our knowledge about the relationship between action and language, and how Broca's area evolved to support them, remains incomplete. A longstanding debate persists regarding whether language and action share the same neural basis, with 2 opposing views. One view proposes that language emerged from action expressed in communicative gestures, and thus they share a common basis [5–7]. The other view sees language as a cognitive ability independent of action [8,9].

Both views—favoring and opposing a shared basis for language and action—built their arguments on theoretical and empirical grounds [5–7]. At the theoretical level, the debate focuses on the (di)similarity between the structure of goal-directed sequential actions and

chimpanzee-broca. An archived version of this data can be found through Zenodo: https://zenodo.org/record/8154437.

**Funding:** This study was funded by the Max Planck Society under the inter-institutional funds of the president for the project "Evolution of Brain Connectivity (EBC)" to AF. This work was supported, in part, by NIH grants AG-067419, NS-092988, and NS- 42867 to WDH and CCS. All aspects of this research conformed to existing US and NIH federal policies on the ethical use of chimpanzees in research. The funders had no role in study design, data collection and analysis, decision to publish, or preparation of the manuscript.

**Competing interests:** The authors have declared that no competing interests exist.

syntax (i.e., the rules that govern how words are arranged in a sentence). While some argue that actions rely on a hierarchical structure of subgoals similar to that of linguistic syntax [10,11], others claim that such a description does not meet the definition of syntactic hierarchy in human language [8]. Meanwhile, at the empirical level, several studies in humans have found action to recruit Broca's area [12,13], an area primarily related to language, thus suggesting a functional codependence between action and language [5,14,15]. However, these studies did not directly compare action against syntactic aspects of language, thus making it hard to understand if the same regions activate for both processes. In this way, the debate concerning action and language is, at its center, about the relationship between action and the core aspect of language, syntax. Other aspects of language (e.g., semantics, phonology) rely on more widely distributed neural networks [16].

To date, only 3 studies directly compared the neural underpinning of action and syntactic aspects of language in humans. Two are meta-analyses, comparing peak activations of syntactic tasks against motor-related ones [17], and syntactic processing with tool use [18]. The third study uses functional imaging to compare syntactic processing with tool use in a within-subject design [19]. All these studies found that language and action recruit largely nonoverlapping areas of Broca's area, with language being processed more anterior than action. In addition, meta-analysis showed that Brodmann Area 44 (BA44), the cytoarchitectonic defined posterior division of Broca's area [20,21], is functionally divided in 2 regions, with language recruiting its anterior part and action recruiting its posterior part [17]. Importantly, this functional subdivision mirrors the underlying distribution of neurotransmitter receptors in BA44, which are a powerful indicator of functional diversity [22].

To help settle the debate on the language/action relationship, we can turn to our close evolutionary relatives [23]. Anthropoid primates, such as chimpanzees and macaques, possess a cytoarchitectonically similar Broca's area homolog that, as in humans, functionally responds to action [4]. Moreover, there is evidence that great apes can master some aspects of language using augmentative or alternative communication systems such as gestures or visual graphic symbols [24]. However, only humans possess the faculty of creating complex multiword utterances following a syntactic hierarchy [25]. Hence, a cross-species comparison between the human linguistic brain and that of one of our close living relatives, the chimpanzees, may shed light on the neural basis of action and language.

Earlier cross-species comparisons have shown that the prefrontal cortex is a region that allometrically scales to increase at a disproportionate rate across primates [23], leading to a relatively large size in the human brain [26,27]. A comparison of the cytoarchitectonic structure of Broca's area in human and macaque brains revealed an enlargement of BA44 and BA45, in particular for the posterior part of BA45 [28]. Although the comparison with macaques is of interest, it has been argued that research focused on our nearest extant relatives, bonobos and chimpanzees, is most relevant to determine which unique features have coevolved with language abilities [23]. Comparing humans and chimpanzees, it was found that the cytoarchitectural regions BA44 and BA45 were up to 6.6-fold larger in humans than in chimpanzees (1.3-fold and 1.4-fold larger than expected, respectively, after correcting for overall cortical enlargement) [29]. Furthermore, based on histological studies, it has been shown that Broca's subregions BA44 and BA45 differ between humans and chimpanzees in terms of their asymmetry. Human BA45 reaches its leftward volumetric asymmetry by the age of 5 years during development. Human BA44 only reaches its asymmetry by the age of 11 years [21] when children acquire full proficiency in semantic and syntactic knowledge [30]. In contrast, in chimpanzees, neither BA44 nor BA45 develops volumetric asymmetry [29].

In the present study, we examined the phylogenetic changes of Broca's area by comparing cytoarchitectural segmentations of BA44 and BA45 in humans and chimpanzees, derived from

published histological data [21,29]. Leveraging advanced cortical registration methods [31–33], we aligned the brains of chimpanzee and human, enabling us to perform a direct comparison of the segmentations across species. Our analysis confirms that Broca's area expanded in humans, with left BA44 being the subregion that enlarged the most. Furthermore, we show that the chimpanzee left BA44 maps to the posterior section of human BA44, a region functionally related to action, having virtually no overlap with the anterior syntax regions. Our results suggest that BA44 evolved from an action region, as found in our close living ape relatives, to a bipartite system with a posterior section recruiting action, and an independent anterior section for syntax. These findings contribute important insights regarding the longstanding debate on the (in)dependence of language and action and the evolution of Broca's area.

## Results

### Symmetry of Broca's area homolog in chimpanzee and a surface probabilistic atlas

Through a semisupervised pipeline (summarized in Fig 1A), we precisely reconstructed the cortical surface of 9 chimpanzee brains from their structural MRI data. Having their surface representation, we projected both BA44 and BA45 volumetric histological segmentations [29] to each individual surface. We examined all individuals for evidence of surface area asymmetry of both histologically defined regions (BA44 and BA45) using a Wilcoxon signed-ranks test. Although there was considerable asymmetry in some individuals (see Fig 1B), both BA44 and BA45 showed no asymmetry at the population level (BA44 T = 16, $p$ = 0.49; BA45 T = 10, $p$ = 0.16).

To enable comparison across subjects, we coregistered the individual brain surfaces to the surface reconstruction of the JUNA [34] chimpanzee template (see Fig 1A). On the JUNA surface, we averaged all the individual segmentations, deriving a high-quality probabilistic atlas of BA44 and BA45 homologs in the chimpanzee brain (Fig 1B). The resulting atlas is open access and available for direct download (see Data and Code Availability Statement).

### Comparison between human Broca's area and its chimpanzee homolog

Leveraging advanced surface registration [31,33], we coregistered the JUNA surface to the surface reconstruction of the MNI-2009c human template (Fig 2A) [35]. This enabled us to compare the human BA44 and BA45 histological atlases derived by Amunts and colleagues [21], with our probabilistic atlas of the chimpanzee homolog (Fig 2B).

After projecting the chimpanzee segmentations to the human brain, we computed their volumes using the MNI template's cortical thickness. We found the projected chimpanzee BA44 to have an average size of 2,331 mm$^3$ (SD: 789) in the left hemisphere, and 1,955 mm$^3$ (SD: 935) in the right hemisphere. In contrast, Amunts and colleagues [21] reported the human BA44 to have an average size of 3,839 mm$^3$ (SD: 2,277) in the left hemisphere, and 2,527 mm$^3$ (SD: 1,597) in the right hemisphere. This means that, when scaled and projected to the same surface template, the human BA44 is 1.64 times larger in the left hemisphere than in the chimpanzee, and 1.29 times larger in the right hemisphere. Moreover, Fig 2B shows that such enlargement is likely the result of a substantial anterior expansion, not present in the right BA44.

For the chimpanzee BA45, the average size after projecting to the human brain (Fig 2B) was 3,187 mm$^3$ (SD: 1,002) and 2,329 mm$^3$ (SD: 1,308) for the left and right hemispheres, respectively. For the same region in humans, Amunts and colleagues [21] reported an average size of 3,242 mm$^3$ (SD: 1,149) and 3,173 mm$^3$ (SD: 1,637) for the left and right hemispheres,

## a. Chimpanzee Reconstruction Pipeline

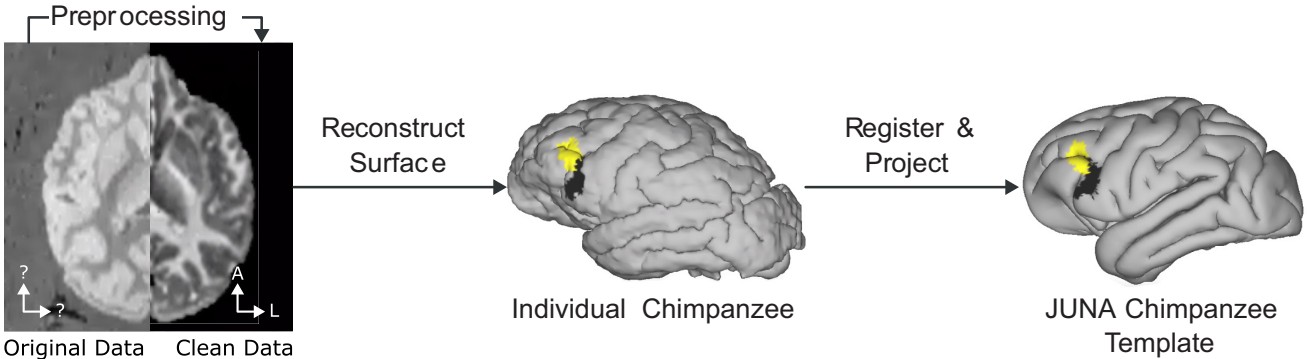

## b. Broca's Homolog in the Chimpanzee: Probabilistic Atlas and Lateralization

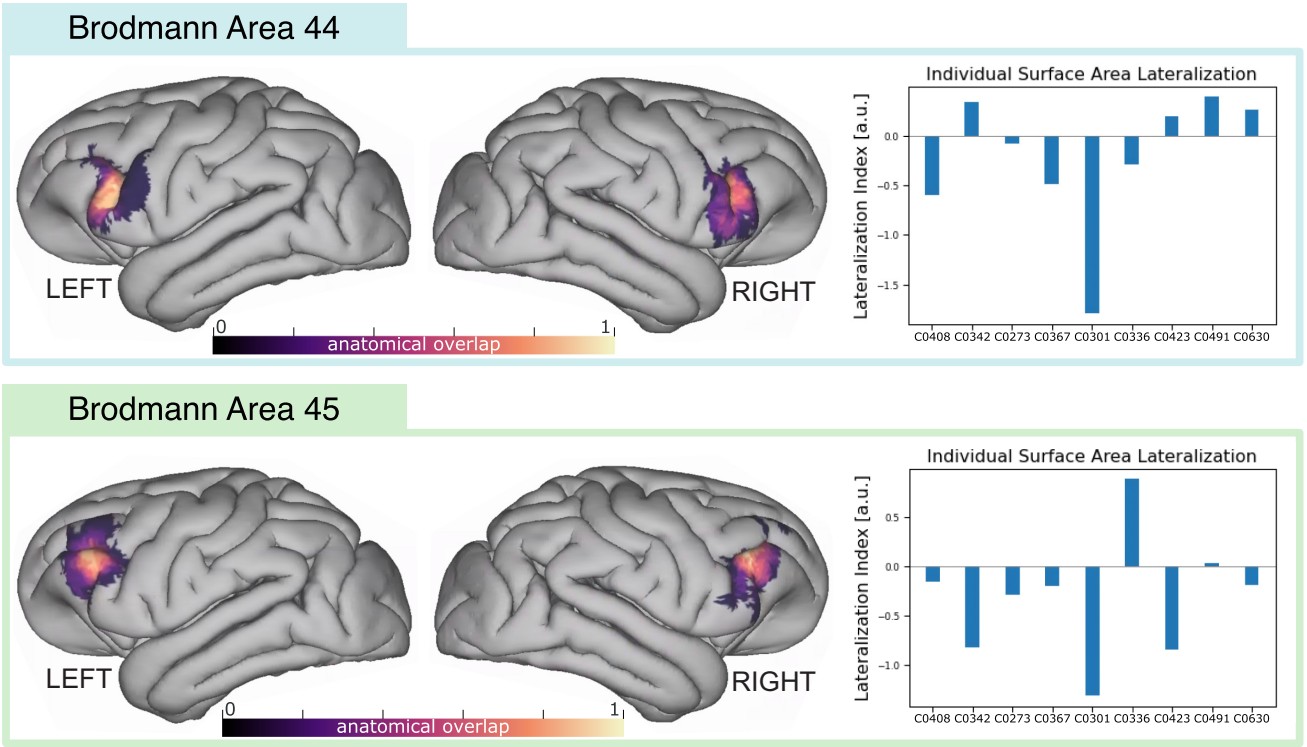

**Fig 1.** **(A)** Reconstruction pipeline for the cytoarchitectonic surface maps. First, the raw MRI data were cleaned using noise reduction and contrast inversion. Next, the individual surfaces were reconstructed in FreeSurfer. The individual maps of BA44 and BA45 are displayed in black and yellow, respectively. Finally, the individual surfaces and cytoarchitectural maps were registered to the JUNA template surface **(B)** Probabilistic atlas of regions BA44 and BA45 in the chimpanzee brain, derived from the individual maps, alongside the lateralization index for each individual brain. The underlying data can be found in the GitHub/Zotero repository, under the results/chimpanzee-atlas folder.

respectively. The human BA45 was only 1.02 times larger in the left hemisphere than the chimpanzee's homolog area, while being 1.36 times larger in the right hemisphere.

### Comparing the projection of chimpanzee BA44 with human functional maps

To better understand the behavioral role of the observed expansion, we computed the overlap of the projected chimpanzee BA44 with functional subdivisions of human BA44 related to

### a. Two Step Surface Registration to Align the Human and Chimpanzee Brain

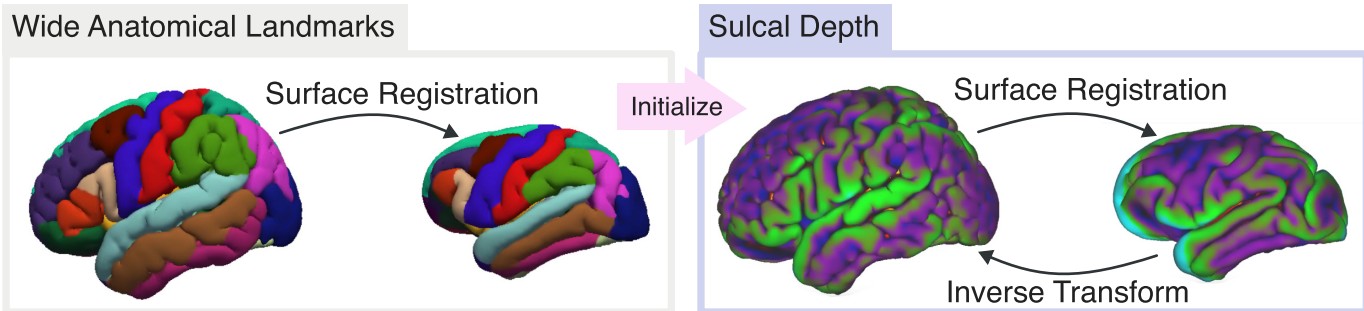

### b. Broca's Cythoarchitectural Atlas - Human vs Chimpanzee Comparison

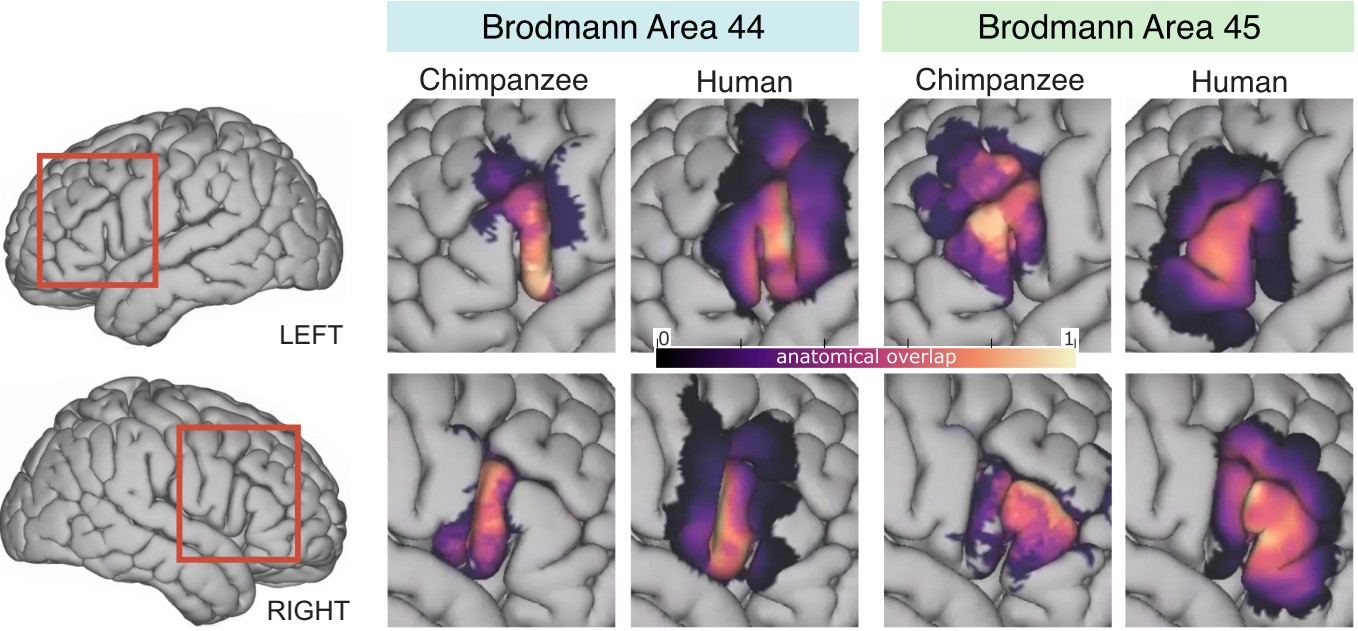

**Fig 2.** **(A)** Two-step surface registration; in the first step, we align gross anatomical landmarks. This first alignment is then used to start a more granular one, based on sulcal depth. **(B)** Side-by-side comparison of our chimpanzee probabilistic atlas with the human population overlap of Amunts and colleagues [21] in the human brain template. Left BA44 is the area that grew the most and shows a large anterior expansion, which is not present in right BA44. The underlying data and scripts used can be found in the GitHub/Zotero repository, under the scripts/ and results/human-comparison folders.

action and syntax [17,36–38]. To compare only with the core chimpanzee BA44, we thresholded the projected atlas at the 0.5 level. We found that the chimpanzee BA44 overlapped most with the regions involved in action [17,36] (Figs 3, left, and S1 and Table 1). The highest overlap was found with the area Clos 4, associated with action imagination [36], of which 34% was contained by the chimpanzee's BA44. Following this were the regions Clos 1 (26% contained, associated with phonology and overt speech tasks [36]), Clos 5 (20%, associated with phonology and semantics [36]), Papitto's region (18%, associated with action execution/imitation [17]), and Clos 2 (7%, associated with semantics, orthography, and covert speech [36]). In contrast, the region Clos 3, associated with basic syntactic operations [36–38], had only a 3% overlap with the chimpanzee BA44. Similar results were obtained when comparing across different levels of thresholding (see S1 Fig). Finally, when visually compared, the projected

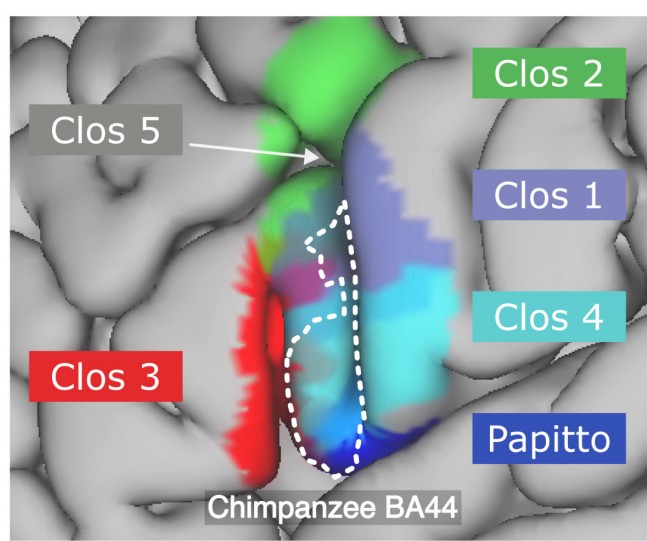

**Fig 3. Percentage of overlap between the chimpanzee BA44 and functional subdivisions of the human BA44 [17,36–38].** Action-related regions present the highest overlap with action-related areas and virtually no overlap with the syntax area. The chimpanzee BA44 atlas was thresholded at 0.5 to maintain only its core area. The functions being reported are those with the highest P (Activation | Domain) as reported by Clos and colleagues [36] and Papitto and colleagues [17], except Clos 1, which was originally reported to be a syntax area, but further studies did not find to be involved in basic syntactic operations [36,37]. The underlying data and scripts used can be found in the GitHub/Zotero repository, under the scripts/ and results/human-comparison folders.

chimpanzee BA44 shared spatial location and extension with the receptorarchitectural division of the human BA44 [22,23] (see S2 Fig).

## Discussion

A longstanding debate persists concerning the relationship between language and action and whether they share a common neural basis. At the center of the debate is the relationship between action and a core aspect of language, syntax. Arguments favoring and opposing their (in)dependence exist based on the structure of action and syntactic processes and the involvement of Broca's area in both abilities. To shed light on the debate, we turned to one of our closest relatives, the chimpanzee, who have a Broca's area homolog involved in action [4] but lack a complex syntactic language. Using robust algorithms [31,33], we aligned human and

**Table 1. Anatomical overlap between functional areas [17,36] with the projected BA44 chimpanzee thresholded at 0.5.** The functional role of each area, as reported by their authors, is stated in the third column. For Clos regions, we report the functions with the highest P (Activation | Domain).

| Region | Area Contained by BA44 | Associated Functions |
| --- | --- | --- |
| Clos 4 | 34% | Action Imagination |
| Clos 1 | 26% | Phonology, Syntax* |
| Clos 5 | 20% | Phonology, Semantics |
| Papitto | 18% | Action Execution / Imitation |
| Clos 2 | 7% | Orthography, Working Memory |
| Clos 3 | 3% | Syntax, Phonology |

*It is important to notice that, even though Clos was originally reported to be a syntax area, further studies did not find it to be involved in basic syntactic operations [37,38].

chimpanzee brains, facilitating a direct comparison of Broca's area cytoarchitectonic subdivisions (BA44 and BA45) [21,29]. We assessed between-species differences in terms of size, symmetry, and spatial location. Furthermore, we leveraged human studies focused on action and syntactic processing to better understand the functional impact of these differences.

### BA44 became increasingly left lateralized in evolution

We tested for asymmetry in surface area for both BA44 and BA45 in chimpanzee brains. Consistent with Schenker and colleagues' [29] volumetric analysis, we found both regions to show no statistical difference in size, thus being symmetric at the population level. This result is in clear contrast with the strongly left lateralized BA44 in humans [21]. Since both chimpanzee and human segmentation were obtained through similar histological procedures, the results support the conclusion that the asymmetry of BA44 developed after humans diverged from our last common ancestor with chimpanzees.

### BA44 expanded the most in humans relative to chimpanzees, extending anteriorly

We projected the BA44 and BA45 of each chimpanzee to the surface of a human cortical template and computed an average volume using the template's cortical thickness. Assuming the MNI template is representative of the Amunts and colleagues [21] population, their reported volumes for humans are directly comparable with our scaled-up volumes for chimpanzee. Our comparison revealed that BA44 enlarged by a factor of 1.64 and 1.29 in the left and right hemispheres beyond the amount of overall cortical expansion, respectively, in this cross-species comparison. Meanwhile, BA45 enlarged only in the right hemisphere, by a factor of 1.36. Our results show that Broca's area enlargement is remarkable in context of the evolution of human prefrontal cortex size [26,29]. Moreover, our findings suggest that BA44 became left lateralized thanks to a large anterior expansion in the left hemisphere.

### The chimpanzee BA44 projects to human areas related to action, and not syntax

We compared the projected chimpanzee atlas with functional subdivisions of human Broca's area. Our results show that the core chimpanzee BA44 overlapped solely with action-related areas, with the greatest overlap found with regions functionally associated with (in descending order) action imagination, phonology, and action execution/imitation [17,36]. Indeed, we found almost no overlap between the core BA44 chimpanzee homolog and the Broca's subdivision involved in basic syntax operations [36–38]. Our findings were consistent across multiple levels of thresholding for the chimpanzee BA44 probabilistic map. These results indicate that a simple anatomical scale and shift of the chimpanzee Broca's area does not explain the existence of the syntax subregion of Broca's area in the human brain.

### Cross-species differences in BA44 support a segregation of action and syntax in humans

Recent functional imaging studies found that both language and action recruit nonoverlapping subdivisions of Broca's area in the human brain, with language being processed more anteriorly than action [17–19,36–38]. Moreover, it has been found that left BA44 segregates action and syntactic processes of language in 2 distinct subregions, with syntax recruiting its anterior part and action the posterior one [17,36–38]. This functional subdivision of human BA 44

mirrors the underlying receptorarchitectonic organization, which is a powerful indicator of functional diversity [22].

In this study, we have provided further evidence for this action/language segregation, finding that it is likely the result of an evolutionary process involving Broca's area. By coregistering chimpanzee and human brains, we found that BA44 underwent a large expansion and left lateralization. Moreover, we found the chimpanzee BA44 maps anatomically to the posterior section of human BA44, functionally associated to action. Indeed, virtually no overlap was found between the chimpanzee BA44 and the human syntax regions. Furthermore, when visually compared, the projected chimpanzee BA44 shared spatial location and extension with the receptorarchitectonic division of human BA44. Taken together, this suggests that the left human BA44 evolved to accommodate syntax through an anterior expansion in the inferior frontal cortex.

Along with contributing to the debate on the relationship between action and language, our findings add to the broader topic of language origins. Although there is an evolutionary continuity in auditory-vocal processing and their underlying neurobiological substrate in temporo-frontal networks [39–42], functional neuroanatomical changes appear to play a crucial role during the evolution of prefrontal cortex. Functional studies indicate that when macaques process simple grammatical sequences activation of both Broca's area and the frontal operculum is observed [43]. In contrast, human brains solely recruit the frontal operculum for simple grammars, whereas BA44 comes into play when processing complex grammatical sequences that nonhuman primates cannot process [44,45]. Thus, despite the observed similarity in the organizational principles across primates, it may have been the expansion of cytoarchitectonically defined BA44 throughout evolution that paved the way for the representation of language in the human brain.

When comparing human and chimpanzee Broca's area, an implicit assumption is that the chimpanzee homolog can serve as a referential model to that of our shared last common ancestor. It is important to acknowledge that this is not entirely accurate, as chimpanzee brains have also certainly changed along their evolutionary lineage. Nevertheless, the limited fossil evidence from endocranial morphology suggests similarities between extant great apes and early hominins in the region of the inferior frontal gyrus, until a more pronounced "Broca's cap" in the left hemisphere becomes evident from certain crania of the genus *Homo* starting at approximately 1.8 million years ago [46]. Of additional note, the anatomical coregistration method we employed cannot offer a complete answer to the exact underlying evolutionary process that Broca's area underwent (e.g., recycling, neural reuse or cultural reuse [47–49]); for a thorough discussion, please refer to Amunts and colleagues [22]. Finally, our sample size is relatively small, meaning further cross-species comparative studies will be needed on the cytoarchitectonic and receptorarchitectonic organization of BA44 to test our conclusions.

Through a cross-species comparison, our study contributes key insights to Broca's area reorganization and the ongoing debate on the relationship between language and action. Our findings support the interpretation that BA44 was modified from an action area, as found in nonhuman primates, to a bipartite system serving syntax anteriorly and action posteriorly. In this way, our results underline distinct neural bases for action and syntactic processes in the human brain, and thus, an independence of both cognitive domains.

## Materials and methods

### Ethics statement

All subjects in this study were housed in accordance with federal and state laws governing the welfare and care of nonhuman primates in the United States. All procedures were approved by

the Emory University Institutional Animal Care and Use Committee (protocol #YER2000673012513). We emphasize that the collection of postmortem brains used in this study were obtained opportunistically when individuals died from natural causes or were euthanized for quality-of-life reasons.

### Cytoarchitecture segmentation of Broca's area in human brains

We downloaded the publicly available data from Amunts and colleagues [21], in which the left and right BA44 and BA45 were manually segmented on 10 subjects following histological procedures. While the individual maps are not available, the Julich institute has released the probabilistic cytoarchitectural map of both areas derived from the Amunts' dataset.

### Cytoarchitecture segmentation of Broca's area homolog in chimpanzee brains

Our chimpanzee cytoarchitectural data come from a previous study [29], in which both BA44 and BA45 were bilaterally delineated and guided by the same cytoarchitectonic criteria used in the human maps [21]. Whole-brain MRI data were acquired ex vivo for all chimpanzees (see S1 Supplementary Methods). From the population of 12 chimpanzees, we discarded 3 based on inadequate corresponding MRI data quality, retaining 9 subjects (*Pan troglodytes*, 5/4 males/females, age = 32.8 ± 11.8 years, age range = 12 to 44.5 years). The demographics of the included chimpanzees are summarized in S1 Table. For additional information on the data acquisition, please refer to the original publication [29].

### Broca's homolog in chimpanzees: Deriving a probabilistic atlas and studying population symmetry

We derived a probabilistic atlas of Broca's area homolog from the individual cytoarchitectonic segmentations of BA44 and BA45 and their associated ex vivo MRI scans. We performed all the analysis on the reconstructed cortical surfaces, as surface analysis better captures and aligns brains based on their gyrification, thus being more robust than volumetric analysis [50].

The procedure can be summarized in 5 steps: (I) reconstruct 3D brain surfaces from the ex vivo MRI scans using FreeSurfer; (II) project each chimpanzee's BA44 and BA45 volumetric segmentation to their corresponding surfaces; (III) register all surfaces to a common template, namely, the JUNA chimpanzee brain template [34]; (IV) map the individual cytoarchitectural regions to the JUNA template; and (V) aggregate them to derive a high-quality probabilistic atlas. See Fig 1A for a graphical explanation and S1 Supplementary Methods for a detailed explanation of each step. The processing scripts for the computation of the open access atlas are readily available for download (see Data and Code Availability Statement).

We further leveraged the individual reconstructions to study the surface-area asymmetry of Broca's homolog in chimpanzee brains. For this, we computed the areas of BA44 and BA45 on each individual chimpanzee and tested their bilateral symmetry through a Wilcoxon signed-ranks test.

### Mapping chimpanzee cytoarchitectural maps to the human brain

To enable cross-species comparison, we aligned the cortical reconstruction of JUNA template [34] to that of the human MNI template (ICBM152 9c Asymmetric) [35]. Given the differences in brain shape and volume, we opted to use surface-based registration algorithms, which have been proven successful in aligning the brains of chimpanzees and humans [33].

Based on the work of Eichert and colleagues [33], we performed the surface-based registration in 2 stages. In the first stage, we performed a first alignment of the brain templates using gross anatomical regions. Specifically, we aligned the brains based on their inferior frontal gyrus, as defined by the Desikan atlas (Fig 2A) [51]. Starting from that rough alignment, we then carried a more granular registration based on the sulcal patterns. For a detailed explanation of each stage, please refer to S1 Supplementary Methods as well as the open access processing script (see Data and Code Availability Statement).

## Expansion of BA44 and BA45 in humans relative to chimpanzees

In their histological study, Amunts and colleagues [21] report the average gray matter volume for human BA44 and BA45. Since our chimpanzee regions stem from similar histological procedures, we can study how much BA44 and BA45 expanded through evolution by mapping them to a common space and by comparing their size across species.

Having morphed chimpanzee BA44 and BA45 to the human template, we computed their individual volumes using the MNI template cortical thickness. In this way, we obtained volumes for chimpanzee Broca's area subregions that are scaled up and projected onto the template human cortical surface. Assuming the MNI template is representative of the Amunts and colleagues [21] population, their reported volumes for humans are directly comparable with our scaled-up volumes for chimpanzee.

## Functional aspects of the BA44 homolog in the human brain

We aimed to understand the relation between function and the location of the projected chimpanzee Broca's area homolog—with a particular interest in language and action. For this, we projected the functional subdivisions of human BA44 defined by Papitto and colleagues [17] and Clos and colleagues [36] to the MNI cortical surfaces. There, we compared them to the core chimpanzee BA44, obtained by thresholding the atlas at the 0.5 level, i.e., the points in the surface where the majority of the chimpanzee population had their BA44 located. Particularly, for each functional region, we computed their overlap with the chimpanzee BA44, defined as how much of the functional area was contained by the chimpanzee BA44. We further visually compared our projected chimpanzee BA44 with an existing receptorarchitectonic division of BA44 [23]. The comparison had to be carried out visually since no volumetric nor surface data are publicly available.

## Supporting information

**S1 Table. Demographics of the included subjects.**
(DOCX)

**S1 Fig. Comparing the overlap of Functional Subdivisions of the Human BA44 and the Chimpanzee BA44 Probabilistic Atlas at different levels of threshold.** Notice that only the probabilistic atlas is being thresholded. As expected, the overlap decreases as the area of BA44 shrinks (the threshold increases). For all threshold levels, the least overlapping region is the syntax-related area Clos 3. The data and plotting script can be found on the GitHub/Zotero repository, under the scripts/human-space folder.
(EPS)

**S2 Fig. Visual comparison between our derived atlases for BA44 and the receptorarchitectonic parcellation from Amunts and colleagues [22].** Left: Receptorarchitectonic areas projected to the lateral surface of an individual postmortem brain as depicted in Amunts and colleagues [22]. Right: Human and chimpanzee BA44 [21,29] projected into the human

template surface. By visual comparison, 44v shares location and extension with the projected chimpanzee BA44. No volumetric or surface data are publicly available, for which only a visual comparison is possible. Please notice that the diagonal sulcus (ds) from the individual brain (Left) corresponds to the ascending branch of the Sylvian fissure in the MNI template (Right) as shown by Sprung-Much and Petrides [52]. The human and chimpanzee data can be found in the GitHub/Zotero repository, under the results/human-comparison folder.
(EPS)

**S1 Supplementary Methods. Chimpanzee data.**
(DOCX)

## Acknowledgments

We thank Hannah Gerbeth for her help during the FreeSurfer white matter segmentation.

## Author Contributions

**Conceptualization:** Cornelius Eichner, Chet C. Sherwood, William D. Hopkins, Alfred Anwander, Angela D. Friederici.

**Data curation:** Guillermo Gallardo, Chet C. Sherwood, William D. Hopkins.

**Formal analysis:** Guillermo Gallardo.

**Funding acquisition:** Chet C. Sherwood, William D. Hopkins.

**Investigation:** Guillermo Gallardo.

**Methodology:** Guillermo Gallardo, Cornelius Eichner, Alfred Anwander, Angela D. Friederici.

**Validation:** Cornelius Eichner.

**Visualization:** Guillermo Gallardo.

**Writing – original draft:** Guillermo Gallardo, Cornelius Eichner, Chet C. Sherwood, William D. Hopkins, Alfred Anwander, Angela D. Friederici.

**Writing – review & editing:** Guillermo Gallardo, Cornelius Eichner, Chet C. Sherwood, William D. Hopkins, Alfred Anwander, Angela D. Friederici.

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
