## [Editor Report · Decision Letter 0]

7 Feb 2023

Dear Dr Gallardo, 

Thank you for submitting your manuscript entitled "Uncovering the Morphological Evolution of Language-Relevant Brain Areas" for consideration as a Short Reports by PLOS Biology.

Your manuscript has now been evaluated by the PLOS Biology editorial staff, as well two academic editors with relevant expertise. I am writing to let you know that we would like to send your submission out for external peer review, and apologize for the delay in getting this feedback to you. One of our academic editors came down with a cold which delayed our discussion a bit.

IMPORTANT: Before we can send your manuscript to reviewers, we need you to complete your submission by providing the metadata that is required for full assessment. To this end, please login to Editorial Manager where you will find the paper in the 'Submissions Needing Revisions' folder on your homepage. Please click 'Revise Submission' from the Action Links and complete all additional questions in the submission questionnaire. At this point you will also be offered the opportunity to provide reviewer suggestions. These are not required, but are much appreciated by the editorial staff. You are also permitted to exclude up to three researchers.

Once your full submission is complete, your paper will undergo a series of checks in preparation for peer review. After your manuscript has passed the checks it will be sent out for review. To provide the metadata for your submission, please Login to Editorial Manager (https://www.editorialmanager.com/pbiology) within two working days, i.e. by Feb 09 2023 11:59PM.

Kind regards,

Kris

Kris Dickson, Ph.D., (she/her)

Neurosciences Senior Editor/Section Manager

PLOS Biology

kdickson@plos.org

---

## [Decision Letter · Decision Letter 1]

29 Mar 2023

Dear Dr. Gallardo,

Thank you for your patience while your manuscript "Uncovering the Morphological Evolution of Language-Relevant Brain Areas" was peer-reviewed at PLOS Biology. It has now been evaluated by the PLOS Biology editors, an Academic Editor with relevant expertise, and by several independent reviewers. 

In light of the reviews, which you will find at the end of this email, we would like to invite you to revise the work to thoroughly address the reviewers' reports.

As you will see, all the reviewers agree that you should tone down the strength of your claims and better develop your arguments (also incorporating additional monkey data/findings). We think that it is important that you address the comments of reviewer #3 who highlights some key issues in your conclusions related to i) potentially secondary loss of the division of BA44 concerned with syntax (at least as an alternative evolutionary scenario to be discussed), but critically ii) the fact that, since chimpanzees do not have language-related areas to map (given they lack language per se) it is not really possible to find any other result than chimpanzee action-related brain areas mapping onto human brain areas. Specifically, as also noted by reviewer #3 we think it is also important to discuss what potential result could lead to the opposite conclusion than that you currently reached. Please also address the rest of the reviewers' issues.

Given the extent of revision needed, we cannot make a decision about publication until we have seen the revised manuscript and your response to the reviewers' comments. Your revised manuscript is likely to be sent for further evaluation by all or a subset of the reviewers.

**IMPORTANT - SUBMITTING YOUR REVISION**

*Re-submission Checklist*

*Published Peer Review*

*PLOS Data Policy*

*Blot and Gel Data Policy*

Sincerely,

Paula

---

Senior Editor

PLOS Biology

REVIEWS:

Reviewer #1: This is very interesting Report, focusing on a key question in the evolution of (neurobiological bases of) language (the relationship between action and syntax), and on an area of the brain that is central to the language circuit ("Broca's region", and in particular BA44). 

The authors rely on histological data and cortical registration methods to offer a close comparison of the morphology of BA44 and 45 between humans and chimpanzees

Their main result is that "the left human BA44 evolved to accommodate syntax through an anterior expansion of action-related regions in the inferior frontal cortex"

This is a claim that is sure to trigger a lot of discussion. The evidence provided will likely be closely examined by proponents of different scenarios for the evolution of syntax. It constitutes a genuine novel finding, in my opnion. I am therefore happy to recommend publication.

My only concern pertains to the rather generic use of the term "syntax" throughout the manuscript. As the authors are aware, "syntax" likely consists o multiple subcomponents/traits, and these are likely to be distributed across the brain. This, of course, does not diminish the role of BA44 is core aspects of syntax, but it may require a bit more qualification. 

I also think that the results presented in this Report are unlikely to "provide a solution for the long- standing debate concerning the structural and functional evolution of Broca's area and its role in action and language". Proponents of the view that syntax evolved out of an Action Grammar capacity are likely to take these results to be evidence for a duplication-and-divergence model (evolutionary continuity as opposed to 'novelty'). I leave it to the authors to decide if their claim should be qualified. I don't feel I should insist on this modification. I think that the results are worthy of publicaion in PLoS Biology as they are, and whatever conceptual position the authors adopt will be adequate, as it sure will fuel discussion in the future. 

Reviewer #2: In their work, Gallardo and colleagues examine the evolution of Broca's area by comparing the cytoarchitectonic segmentations of BA44 and BA45 in humans and chimpanzees.

They reconstructed the cortical surface of nine chimpanzee brains using MRI data and aligned brains from chimpanzees and humans to perform a direct comparison of segmentations between the species. They found no BA45 asymmetry at chimpanzee level unlike for humans. Furthermore, they showed an expansion of Broca's areas in humans, with the left BA44 being the most enlarged. The authors conclude that BA44 in humans has evolved from a purely action-related region to a more comprehensive region, with a posterior region that supports action and an anterior region that processes syntax.

The results are novel and will potentially advance our understanding of the evolution of Broca's area in the primate lineage. The study is methodologically sound, and the number of animals used is sufficient for a study with non-human primates. I have only one major concern regarding the overall approach that the authors should consider in their revision.

1. The authors strongly suggest that their data will help improve our understanding of the evolution of Broca's area in humans. Here, they argue that Broca's area may have evolved continuously from precursor structures in the primate lineage. However, to be truly convincing here, it would be essential to go back another step in evolution and look at the morphology of the monkey homolog of Broca's area. This is particularly suggested by the fact that there are a large number of studies present showing morphological and physiological similarities between the macaque monkey homolog and the human Broca's area (notably the extensive work by Pandya and Petrides, but also others). In addition, there are already several reviews addressing the evolution of Broca's area in the primate lineage. I strongly recommend that the authors consider the monkey data and incorporate and discuss the existing review articles accordingly.

Reviewer #3: This short report presents a comparative analysis of human BA44/45 and their chimpanzee homologues in which they probe a hypothetical allometric expansion of these areas in humans vs chimpanzees. The principal result is that the anterior division of the area referred to as "Broca's region" in humans shows a 'disproportionately' great expansion vis-a-vis the right homologue and the more posterior subregion. The methods appear solid, the data and methods are available. The statistical analyses are acceptable. The findings are novel but their importance currently seems over-stated. The rationale for the study is that there are two opposing views on the emergence of language, either from action or independently of it. A key reference is not cited: Zilles & Amunts (2018) discuss comparisons between human and non-human primate LIFG anatomy and this work should be considered, and possibly foregrounded (especially given that it explicitly calls for a comparison between human and chimpanzee neuroanatomy) in the introduction.

They authors make the bold claim that this:

 i. Shows that BA44 "evolved from a purely action-related region"

 ii. Solves a longstanding debate regarding the structural and functional evolution of Broca's area and its role in action and language

It is unclear how looking at anthropoid primates can "settle the debate". The neural basis of action and language in humans is already known (characterised to some extent, even if not completely described), it is not by examining non-human primates that these can be better elucidated.

The assertion that the manuscript provides "a complete picture of Broca's area evolution and … solution to the longstanding debate of BA44's role in language and action" is not well supported. The comparisons presented could not provide a full picture (but nor would they strictly need to), and moreover the function of human BA44 in language and action can only be established by studying humans. Mapping cytoarchitectonic homologues does not itself demonstrate any particular functional role and drawing inferences about the presumed functional drift of a region over evolution requires more convincing argumentation.

If I have understood the argumentation correctly, the authors suggest that because chimpanzee BA44 maps onto only a small, posterior portion of human BA44, human BA44 has undergone an anterior expansion since the LCA with chimpanzees. This seems to depend upon an assumption that the human BA44 is a cytoarchitectonically homogeneous region. Since anterior BA44 (loosely) tends to be found to process syntax, the assumption is that the "syntax part" is derived from the "action part". This result supposedly answers the question of whether the syntax processing region in anterior BA44 evolved from action processing, and not independently. This could be a relatively compelling argument, and certainly fits with a prevailing view (as described int he introduction), but it is need of additional support.

On the basis of the evidence presented it would be equally possible to claim (albeit perversely) that chimpanzee BA44 evolved from a region that supported both syntactic and action processing and lost the division concerned with syntax. It seems important to consider that chimpanzees are not the human/chimpanzee LCA - and that they have presumably also been undergoing evolutionary changes over the past 5-10million years.

Anderson (2010) proposes neural reuse as a mechanism whereby a cytoarchitectonically homologous region of human and non-human cortex could become functionally differentiated. Some kind of argumentation along these lines (or any theory the authors favour) is required to help lend credibility to the proposals.

I feel that there is a flaw in the current set-up of the study, but this may be a misunderstanding arising due to the density of the manuscript. Given the widely accepted assumption that chimpanzees have no language (as defined by the authors), it would be impossible for any result to arise other than one showing that chimpanzee action-related brain areas map onto human brain areas (as there are no chimpanzee language-related areas to map). It would therefore seem impossible for the authors to test the hypothesis that language arose independently of action using this method. Even if Chimpanzee BA44 mapped directly onto human BA44, this would provide evidence for or against - it would merely indicate that some functional modification of a previously homologous region must have taken place. In the present case it is hypothesised that there was an expansion, but if that expansion could only be found to have arisen from a region that is characterised in chimpanzees and therefore non-linguistic, it does not seem that the debate has been solved. If this is not the case, it would be very important to discuss which result would lead to the opposite conclusion. Mapping cytoarchitectonic probability maps from one atlas to another using the methods employed here could never lead to the discovery of an area in one species that does not map onto some region of the other.

Thus, the results regarding the extent of allometric expansion and the asymmetry are themselves compelling, but the interpretation is pushed too far.

To sum up - I am not opposed to the authors' position that language conceivably arises on the basis of action systems. However I do not believe that the present paper demonstrates this as conclusively as claimed. The pattern of allometric expansion from chimpanzee to human with a greater expansion in the left than right hemisphere and a greater expansion in BA44 than 45 is extremely intriguing and these analyses are of interest to the community of researchers concerned with the evolution of the neurobiological bases of language. However the argumentation is under-developed, limitations and fundamental assumptions relevant to the approach are not really discussed and there is no convincing control analysis that can help to support that the pattern of results is related to language specifically. My recommendation is therefore for substantial modification to the introduction and discussion to justify the approach and the strength of the claims.

---

## [Decision Letter · Decision Letter 2]

5 Jul 2023

Dear Dr. Gallardo,

Thank you for your patience while we considered your revised manuscript "Uncovering the Morphological Evolution of Language-Relevant Brain Areas" for publication as a Short Reports at PLOS Biology. This revised version of your manuscript has been evaluated by the PLOS Biology editors, the Academic Editor, and the original reviewers.

Based on the reviews, we are likely to accept this manuscript for publication, provided you satisfactorily address the remaining points raised by the reviewers. Please also make sure to address the following data and other policy-related requests.

1. DATA POLICY:

Regardless of the method selected, please ensure that you provide the individual numerical values that underlie the summary data displayed in the following figure panels as they are essential for readers to assess your analysis and to reproduce it: Figures 1B, 2B, 3, and Supplementary Figures S1, S2.

**Please also ensure that figure legends in your manuscript include information on where the underlying data can be found, and ensure your supplemental data file/s has a legend.**

2. Please note that sole deposition of data or code to GitHub would not be compliant with our policies, as this could be changed after publication (https://journals.plos.org/plosbiology/s/data-availability). However, once the data/code is final, you can archive your publicly available GitHub data to Zenodo. Once you do this, it will also generate a DOI number that you can provide us with. See the process for doing this here: https://docs.github.com/en/repositories/archiving-a-github-repository/referencing-and-citingcontent

3. We suggest a change in the title: "Part of the human language-relevant brain region Broca's area evolved from an action-related area in non-human primates".

We expect to receive your revised manuscript within two weeks.

*Published Peer Review History*

*Press*

Sincerely,

Paula

---

Senior Editor,

pjaureguionieva@plos.org,

PLOS Biology

Reviewer remarks:

Reviewer #1: I am grateful to the authors for taking my comments into consideration. I am particularly pleased with the fact they decided to tone down their conclusions. I endorse publication.

Reviewer #2: The authors have done a good job revising their paper and toning down the impact of some of their findings. However, my major concern has not been adequately addressed.

I agree with the authors that additional studies on cytoarchitecture and connections in apes will help to learn more about the evolution of Broca's area. However, to fully understand the evolution, we also critically rely on neurophysiological data in our closest relatives, and for these invasive experiments we rely on monkeys. Although more distantly related to humans than apes, the similarities are striking, as demonstrated in the extensive work of Petrides and discussed in several reviews in recent years (e.g., Fröhlich et al. 2019 Biol Rev; Rauschecker 2018 Curr Opin Behav Sci; Aboitiz 2018 Front Neurosci; Hage & Nieder 2016 TINS). Moreover, several laboratories have recorded neural activity from ventrolateral prefrontal areas in monkeys showing similar activity patterns before and during vocal output as in human Broca's area before and during human speech signals. Therefore, I would like to emphasize again how important it would be to consider the abundant data on cortical control mechanisms available from the monkey studies, at least in the discussion. This inclusion would increase the impact of the manuscript, which is crucial, especially considering that the paper has lost impact due to the necessary toning down of the results.

Reviewer #3: I enjoyed reading the revised manuscript and feel that the authors' substantial and considered modifications have improved the work. They have addressed my preceding concerns and I wish to express my appreciation of their efforts to address them. I have no further comments.

---

## [Editor Report · Decision Letter 3]

21 Jul 2023

Dear Dr. Gallardo,

Thank you for the submission of your revised Short Reports "Uncovering the Morphological Evolution of Broca’s area: A Comparison of Human and Chimpanzee Brains" for publication in PLOS Biology. On behalf of my colleagues and the Academic Editor, Simon Townsend, I am pleased to say that we can in principle accept your manuscript for publication, provided you address any remaining formatting and reporting issues. These will be detailed in an email you should receive within 2-3 business days from our colleagues in the journal operations team; no action is required from you until then. Please note that we will not be able to formally accept your manuscript and schedule it for publication until you have completed any requested changes.

We suggest a change in the title to improve readability and accessibility: "Morphological evolution of the human language-relevant area of Broca in comparison with the chimpanzee brain" or "Morphological evolution of language-relevant brain areas".

The academic editor also suggests a couple of edits (in capital letters) that you could change along with the title and the requests from the journal operations team:

1. Change "during the evolution of THE prefrontal cortex"

2. Change to: "In contrast, human brains solely recruit the frontal operculum for simple grammars, whereas BA44 comes into play when processing MORE complex grammatical sequences which non-human primates HAVE BEEN ARGUED TO BE unable to process [44,45].

PRESS

Sincerely, 

Paula

---

Senior Editor

PLOS Biology
